# Role of Rifaximin in the Prognosis of Critically Ill Patients with Liver Cirrhosis

**DOI:** 10.3390/antibiotics14030287

**Published:** 2025-03-10

**Authors:** Zhaohui Bai, Congcong Li, Yongjie Lai, Xiaojuan Hu, Luwen Shi, Xiaodong Guan, Yang Xu

**Affiliations:** 1Department of Pharmacy Administration and Clinical Pharmacy, School of Pharmaceutical Sciences, Peking University, Beijing 100191, China; bai_zhao_hui@foxmail.com (Z.B.); laiyongjie@bjmu.edu.cn (Y.L.); shilu@bjmu.edu.cn (L.S.); 2Department of Respiratory and Critical Care Medicine, General Hospital of Northern Theater Command, Shenyang 110840, China; licong1988@hotmail.com; 3Postgraduate College, Shenyang Pharmaceutical University, Shenyang 110016, China; 15841426396@163.com

**Keywords:** liver cirrhosis, rifaximin, critically ill patients, survival

## Abstract

**Background/Objectives**: Critically ill patients with liver cirrhosis impose a substantial health burden on the world. Rifaximin is a potential treatment option for such patients. **Methods**: We extracted critically ill patients with liver cirrhosis from the Medical Information Mart for Intensive Care (MIMIC) IV database. Based on study outcomes, the current study included prevention and treatment cohorts. A 1:1 propensity score matching (PSM) analysis was performed to match the characteristics of patients. The risk of ICU admission and intensive care unit (ICU), in-hospital, 90-day, and 180-day death were explored. Cox regression analyses were conducted, and hazard ratios (HRs) and 95% confidence intervals (CIs) were calculated. Kaplan-Meier curves were further drawn to demonstrate the cumulative 90-day and 180-day survival rate. **Results**: Overall, 5381 critically ill patients with liver cirrhosis were included. In the prevention cohort, rifaximin could decrease the risk of ICU admission (HR = 0.427, 95%CI: 0.338–0.539, *p* < 0.001). In the treatment cohort, rifaximin could decrease the risk of ICU (HR = 0.530, 95%CI: 0.311–0.902, *p* = 0.019) and in-hospital death (HR = 0.119, 95%CI: 0.033–0.429, *p* = 0.001) in critically ill patients with liver cirrhosis. However, rifaximin could not decrease the risk of 90-day (HR = 0.905, 95%CI: 0.658–1.245, *p* = 0.541) and 180-day (HR = 1.043, 95%CI: 0.804–1.353, *p* = 0.751) death in critically ill patients with liver cirrhosis. Kaplan-Meier curve analyses also showed that rifaximin could not significantly decrease the 90-day (*p* = 0.570) and 180-day (*p* = 0.800) cumulative mortality. **Conclusions**: This study suggests that rifaximin can significantly decrease the risk of ICU admission and improve short-term survival but does not impact long-term survival in critically ill patients with liver cirrhosis.

## 1. Introduction

Cirrhosis imposes a substantial health burden on many countries and this burden has steadily increased at the global level since 1990, which will persistently rise in the future [1,2]. Acute decompensation events were the major cause of death in patients with liver cirrhosis [3] and many acute decompensated cirrhotic patients require admission to the intensive care unit (ICU) [4,5]. In the United States, 7–8% of hospitalized cirrhotic patients are admitted into the ICU and the related cost is more than 2 billion dollars per year [6,7].

As well as resuscitation, the management of bacterial infections, which is significantly associated with prognosis [8], is crucial in critically ill patients with cirrhosis [4,5]. Translocation of bacteria and bacterial products transport from gut to circulation through mesenteric lymph nodes and portal vein, which is caused by the liver cirrhosis-associated immune dysfunction and alterations in gut barrier, is the major source of infections in patients with liver cirrhosis [9,10]. Traditionally, third-generation cephalosporins and quinolones were the first-line option for the prevention and treatment of infections in cirrhotic patients [9,11], but this consensus may need to be changed due to the significantly increasing prevalence of gram-positive and multi-drug resistant (MDR) infections [12,13,14].

Rifaximin is a non-aminoglycoside intestinal antibiotic with minimal gastrointestinal absorption and broad-spectrum antibacterial activity covering both gram-positive and gram-negative organisms [15]. Until now, the approved indications of rifaximin by FDA included the treatment of travelers’ diarrhea and irritable bowel syndrome with diarrhea and the prevention of overt hepatic encephalopathy (HE) recurrence [16]. However, in liver cirrhosis, series studies suggest that rifaximin might have other beneficial effects on the course of cirrhosis by modulating the gut microbiome and affecting the gut-liver axis, which include spontaneous bacterial peritonitis (SBP), hepatorenal syndrome (HRS), portal hypertension, and survival, but the quality of evidence is poor [17]. Despite this, rifaximin is still considered a promising disease-modifying agent in decompensated cirrhosis [18]. It should be noted that critically ill patients, especially those admitted to the ICU, commonly rely on treatment with broad-spectrum antibiotics [19]. Thus, in this category, whether rifaximin can provide additional therapeutic benefit when combined with broad-spectrum antibiotics also remains controversial [20,21]. The latest RCT (ARiE trial) showed that antibiotics plus rifaximin can benefit the survival of critically ill decompensated cirrhotic patients with HE, but not in patients with acute chronic liver failure (ACLF) [21].

For this reason, we conducted a retrospective study based on real-world data to further explore the role of rifaximin in critically ill patients with liver cirrhosis.

## 2. Results

### 2.1. Patient Characteristics

Overall, 5381 critically ill patients with liver cirrhosis were included in the current study (Figure 1). Among them, the median age was 59.67 years and 35.2% of patients were female. Over 69% of the patients were White. The median MELD score was 14.74. Regarding comorbidities, 36.10% of patients had hypertension, 30.90% had diabetes, and 16.30% had CKD. The leading causes of liver cirrhosis were viral hepatitis (32.80%) and alcohol hepatitis (24.80%). In terms of complications of liver cirrhosis, the prevalence of ascites, HE, GIB, HRS, HCC, ACLF, and SBP were 35.10%, 7.00%, 2.60%, 5.60%, 2.40%, 2.70%, and 5.00%, respectively. GGT and CRP were excluded from analysis due to missing data greater than 30%. Among the patients, 35.50% had ICU admission records during hospitalization, and the in-hospital mortality was 9.30% (Table 1).

### 2.2. Prevention Cohort

#### Rifaximin and Risk of ICU Admission

A total of 4917 patients were included in the prevention cohort (Figure 1). Among them, 803 patients received rifaximin, while 4114 patients did not (Appendix A). Before PSM, patients in the rifaximin group had significantly more severe critical illness than those in the non-rifaximin group (Appendix A). After PSM, 663 patients were assigned to each group, and the demographic data, baseline data, and treatment data were similar between the rifaximin and no rifaximin groups (Appendix A). Cox regression analysis suggested that rifaximin significantly reduced the risk of ICU admission (Crude HR = 0.421, 95%CI: 0.334–0.531, *p* < 0.001, Adjusted HR = 0.427, 95%CI: 0.338–0.539, *p* < 0.001) in critically ill patients with liver cirrhosis (Table 2). Subgroup analysis showed that patients could significantly benefit from rifaximin regardless of gender, ascites, HE, GIB, HRS, and ACLF, but did not significantly in patients with CKD, SBP or HCC (Figure 2). Additionally, rifaximin, whether combined with or without broad-spectrum antibiotic, was associated with a reduced risk of ICU admission. RCS regression analysis showed that the benefits of rifaximin began when the total dose exceeded 1100 mg (Appendix A).

### 2.3. Treatment Cohort—ICU Sub-Cohort

#### Rifaximin and Risk of ICU Death

A total of 1910 patients were included in the ICU sub-cohort. Among them, 433 patients received rifaximin during their ICU stay, while 1477 patients did not (Appendix A). Before PSM, patients in the rifaximin group had significantly more severe critical illness than those in the non-rifaximin group (Appendix A). After PSM, 228 patients were assigned to each group, and the demographic, baseline, and treatment data were similar between the rifaximin and no rifaximin groups (Appendix A). Cox regression analysis suggested that rifaximin could reduce the risk of ICU death (Crude HR = 0.516, 95%CI: 0.304–0.876, *p* = 0.014, Adjusted HR = 0.530, 95%CI: 0.311–0.902, *p* = 0.019) in critically ill patients with liver cirrhosis (Table 2). Subgroup analysis revealed that male patients and those with ascites or HE significantly benefited from rifaximin. However, no significant benefit was observed in patients with CKD, GIB, HRS, SBP, HCC, or ACLF (Figure 3). Additionally, rifaximin combined with broad-spectrum antibiotics may offer a greater benefit for ICU survival compared to rifaximin alone, although the difference was not statistically significant. RCS regression analysis showed that the benefits of rifaximin began when the total dose exceeded 100 mg, with the optimal dose being 1780 mg. However, when the total dose exceeded 2140 mg, this benefit was lost (Appendix A).

### 2.4. Treatment Cohort—Non-ICU Sub-Cohort

#### Rifaximin and Risk of In-Hospital, 90-Day, and 180-Day Death

A total of 3471 patients were included in the non-ICU sub-cohort. Among them, 660 patients received rifaximin during hospitalization, while 2811 patients did not (Appendix A). Before PSM, patients in the rifaximin group had significantly more severe critical illness compared to those in the non-rifaximin group (Appendix A). After PSM, 493 patients were assigned to each group, and the demographic, baseline, and treatment data were similar between the rifaximin and no rifaximin groups (Appendix A).

*In-hospital death*. Cox regression analysis suggested that rifaximin could significantly reduce the risk of in-hospital death (Crude HR = 0.141, 95%CI: 0.040–0.494, *p* = 0.002, Adjusted HR = 0.119, 95%CI: 0.033–0.429, *p* = 0.001) in critically ill patients with liver cirrhosis (Table 2). Subgroup analysis showed that male patients and those with ascites significantly benefited from rifaximin (Appendix A). Whether rifaximin was combined with broad-spectrum antibiotics or not, it still significantly reduced the risk of in-hospital death. RCS regression analysis showed that the benefits of rifaximin began when the total dose exceeded 100 mg, with the optimal total dose being 1000 mg. However, when the total dose exceeded 1770 mg, this benefit was lost (Appendix A).

*90-day and 180-day death.* Cox regression analysis suggested that rifaximin could not reduce the risk of 90-day (Crude HR = 0.912, 95%CI: 0.664–1.253, *p* = 0.571, Adjusted HR = 0.905, 95%CI: 0.658–1.245, *p* = 0.541) and 180-day (Crude HR = 1.035, 95%CI: 0.798–1.342, *p* = 0.796, Adjusted HR = 1.043, 95%CI: 0.804–1.353, *p* = 0.751) death in critically ill patients with liver cirrhosis (Table 2). Kaplan-Meier curve analyses also showed that rifaximin could not significantly increase the 90-day (Figure 4A) (Log-rank test: *p* = 0.570) and 180-day (Figure 4B) (Log-rank test: *p* = 0.800) cumulative survival rate of critically ill patients with liver cirrhosis. Subgroup analysis showed that patients did not significantly benefit from rifaximin in any subgroup (Appendix A). RCS regression analyses showed that rifaximin did not significantly benefit 90-day or 180-day survival at any total dose (Appendix A).

## 3. Discussion

In this retrospective cohort study, we systematically evaluated the effect of rifaximin on the prevention of ICU admission and the improvement of survival in critically ill patients with liver cirrhosis. There were two major findings: (1) rifaximin significantly reduced the risk of ICU admission in critically ill patients with liver cirrhosis, and (2) rifaximin significantly improved the ICU and in-hospital survival but did not the 90-day and 180-day survival in these patients.

Recently, two randomized controlled trials (RCTs) have investigated the role of rifaximin in critically ill patients with liver cirrhosis [20,21]. Compared to these studies, our current research had several distinguishing features. First, for the population, Kulkarni’s study [21] included patients who were critically ill patients with liver cirrhosis and acute HE, whereas both Ward’s [20] and our studies included all critically ill patients with liver cirrhosis, regardless of HE status. Moreover, while all patients in Ward’s and Kulkarni’s studies were admitted to the ICU, our study also included patients who did not require ICU admission. Second, for the intervention, in Ward’s study [20] all patients initially received rifaximin, with the control group discontinuing the treatment, but in Kulkarni’s [21] and our studies, the control group did not receive rifaximin. Third, for clinical outcomes, Ward’s [20] study focused on the impact of rifaximin on ICU length of stay and ICU mortality and Kulkarni’s [21] study explored the resolution of HE, time to resolution of HE, and in-hospital mortality. In contrast, our study explored the effect of rifaximin on the risk of ICU admission, the risk of ICU, in-hospital, 90-day, and 180-day death. Fourth, our study conducted a more detailed subgroup analysis based on gender, comorbidities, and complications than Ward’s [20] and Kulkarni’s [21] studies. Finally, our study had a larger sample size compared to the previous studies, providing more robust statistical power. Regarding findings, Ward’s study suggested that rifaximin had no effect on ICU mortality or length of stay. Kulkarni’s study found that rifaximin did not improve the resolution of HE or in-hospital mortality in patients with ACLF, although it did benefit the in-hospital mortality in decompensated cirrhosis patients. In contrast, our study found that rifaximin not only decreased the risk of ICU admission but also reduced ICU and in-hospital mortality. However, it did not improve long-term survival (90-day or 180-day mortality). Furthermore, our study confirmed that rifaximin did not benefit the survival of patients with ACLF.

Patients with liver cirrhosis are particularly influenced by infections due to immune dysfunction and intestinal barrier impairment, thereby leading to frequent antibiotic use, especially for third-generation cephalosporins and quinolones, and an associated rise in multidrug-resistant (MDR) organisms in this population. Previous epidemiology studies showed that the prevalence of multidrug-resistant bacterial infections in patients with liver cirrhosis was more than 35% [13,14]. Meanwhile, the drug-resistance of third-generation cephalosporins and quinolones is also rising [22,23,24], highlighting the urgent need for new antibiotic strategies [13]. Rifaximin, a gut-specific antibiotic with minimal systemic absorption, has been approved for the prevention of episodes of HE in patients with liver cirrhosis and is considered to have a low risk of contributing to systemic antibiotic resistance. Several previous studies suggested that rifaximin had many putative effects on the gut-liver axis, which mainly included the increasing of intestinal epithelial homeostasis, inhibition of NF-κB, decreasing of TNF-α, IL-6, IL-8, and IL-10 secretion, and induction of biotransformation phase 1 and 2 enzyme activities. Additionally, rifaximin leads to normalization of serum lipopolysaccharide binding protein levels and thereby lowering of the proinflammatory state in the liver. These mechanisms suggest that rifaximin may offer therapeutic benefits for various complications of liver cirrhosis, especially those involving intestinal bacterial translocation and systemic inflammation [18,25].

Recently, the potential for rifaximin to contribute to daptomycin resistance has raised concerns among researchers [26]. Despite these concerns, the potential benefits of rifaximin in patients with liver cirrhosis should be carefully weighed against the risks. Several studies have pointed out that while current evidence highlights a distinct and significant resistance mechanism, the clinical relevance of this finding remains unclear. As such, there is not yet enough compelling evidence to warrant immediate and fundamental changes in clinical practice [27].

Several limitations should be acknowledged in the current study. First, we were unable to definitively determine the specific indications for rifaximin use in each patient. Second, we could not confirm whether rifaximin was prescribed prior to hospital admission. Third, the study design did not allow us to explore rifaximin-related drug resistance. Fourth, as the study relied on electronic healthcare records from routine clinical practice, there were issues with missing data and outliers. To mitigate bias from missing data, we employed multivariate imputation by chained equations to retain statistical power. Fifth, the MIMIC-IV database spans over a decade, and changes in clinical guidelines for liver cirrhosis during this period could have influenced the relevance of our findings. Sixth, CRP, which is a common parameter for the evaluation of a critically ill patient, was a major missing data point in the baseline information due to the database. Seventh, the cause of admission to the ICU cannot be extracted. Eighth, the efficacy of rifaximin cannot be assessed in the current study. Ninth, during ICU stay, the association between dose of rifaximin and its benefit of survival needs further exploration due to the small sample size in the current study. Eleventh, the timing of broad-spectrum antibiotics association with rifaximin cannot be clarified in the current study. Finally, the use of a single database requires further validation in diverse cohorts to confirm the beneficial effects of rifaximin.

In conclusion, rifaximin administration provided significant benefits for ICU admission and ICU and in-hospital survival of critically ill patients with liver cirrhosis. Excluding HE, rifaximin may benefit other complications of liver cirrhosis. However, prospective interventional trials are needed to confirm these findings in the future.

## 4. Materials and Methods

### 4.1. Data Source and Study Design

We used the Medical Information Mart for Intensive Care (MIMIC) IV database, which is a large, freely available database comprising deidentified health-related data from patients who were admitted to the critical care units of the Beth Israel Deaconess Medical Center [28]. MIMIC-IV includes data from 2008–2019. Zhaohui Bai, as one of the authors, obtained authorization to access the database (Record ID: 40343857). Critically ill patients were included based on the following eligibility criteria: (1) patients with liver cirrhosis who were consecutively admitted to hospital between 2008–2019; and (2) patients with a hospital length of stay greater than 24 h. For patients admitted to the ICU more than once, we analyzed only the first ICU stay.

In the current study, we divided the patients into two cohorts (prevention and treatment cohorts) according to the study outcomes. In the prevention cohort, we explored the effects of rifaximin on the prevention of ICU admission in critically ill patients with liver cirrhosis. Thus, we further excluded patients who were directly admitted to ICU upon admission in the prevention cohort (Figure 1). In the treatment cohort, we further divided patients into two sub-cohorts: ICU and non-ICU. The ICU sub-cohort included patients who had ICU admission records during hospitalization, and we explored the effects of rifaximin on the ICU survival of critically ill patients with liver cirrhosis. The non-ICU sub-cohort included patients who did not have ICU admission records during hospitalization, and we explored the effects of rifaximin on the in-hospital, 90-day, and 180-day survival of critically ill patients with liver cirrhosis.

### 4.2. Variable Extraction and Data Collection

Data extraction was conducted for the patients included as follows:

*Demographic data*: Age, gender, and race.

*Baseline data*: Admission type, time of admission, comorbidity (i.e., hypertension, diabetes, chronic kidney disease (CKD), and infections), etiology of liver cirrhosis (i.e., viral hepatitis, alcohol hepatitis, autoimmune hepatitis, and non-alcoholic fatty liver disease (NAFLD)), complications of cirrhosis (i.e., ascites, HE, gastrointestinal bleeding (GIB), HRS, SBP, ACLF, hepatocellular carcinoma (HCC), and other liver failure), laboratory data (i.e., hemoglobin (Hb), white blood cell (WBC), platelet (PLT), total bilirubin (TBIL), alanine aminotransferase (ALT), aspartate aminotransferase (AST), serum albumin (ALB), alkaline phosphatase (ALP), glutamyl transpeptidase (GGT), serum creatinine (Scr), sodium (Na), potassium (K), prothrombin time (PT), international normalized ratio (INR), and C-reactive protein (CRP)). Model for end-stage liver disease (MELD) score was calculated. For patients in the ICU sub-cohort, the laboratory data were collected as the highest recorded values within 24 h of ICU admission. Additionally, ICU-specific baseline information, such as ICU intime, vital signs (i.e., heart rate, respiratory rate (RR), SpO_2_, anion gap, blood pressure, and temperature) were extracted, and sequential organ failure assessment (SOFA) score was calculated.

*Treatment data*: Liver transplantation, hemodialysis, rifaximin, total dose of rifaximin, broad-spectrum antibiotic, human albumin, and vasoactive agent. For ICU sub-cohort, treatment data before ICU admission and ventilation during ICU admission were also extracted.

*Outcome data*: Discharge time and deathtime. Additionally, for patients in the ICU sub-cohort, ICU out time, and ICU death were also extracted.

Additionally, for laboratory data and vital signs, variables with more than 30% missing data were excluded (Appendix A). For variables with less than 30% missing data, missing values were imputed using the Multivariate imputation by chained equations (MICE) method.

### 4.3. Definitions and Outcomes

We conducted a thorough identification of liver cirrhosis and its complications and comorbidities, based on the ICD-9-CM and ICD-10-CM codes [29,30] (Appendix A). For the prevention cohort, the rifaximin group was defined as follows: (1) patients without ICU records who received rifaximin during hospitalization, and (2) patients with ICU records who received rifaximin before ICU admission; The non-rifaximin group was defined as follows: (1) patients without ICU records who did not receive rifaximin during hospitalization, and (2) patients with ICU records who did not receive rifaximin before ICU admission. For the non-ICU sub-cohort, the rifaximin group was defined as patients receiving rifaximin during hospitalization, otherwise it was the non-rifaximin group. For the ICU sub-cohort, the rifaximin group was defined as patients who received rifaximin during ICU admission, otherwise it was the non-rifaximin group. Broad-spectrum antibiotic therapy was defined according to Ward et al.’s study [20] (Appendix A).

For the prevention cohort, the primary outcome was the rate of ICU admission. For the ICU sub-cohort, the primary outcome was ICU survival. For the non-ICU sub-cohort, the primary outcomes included in-hospital, 90-day, and 180-day survival.

### 4.4. Statistical Analysis

Continuous variables were reported as medians and interquartile ranges (IQRs) and were compared using the non-parametric Mann-Whitney U test. Categorical variables were reported as frequencies (percentages) and were compared using the chi-square test. A 1:1 propensity score matching (PSM) analysis was performed with the matching factors including demographic data (age, gender, and race), baseline data (admission type, MELD score, hypertension, diabetes, CKD, viral hepatitis, alcohol hepatitis, ascites, HE, GIB, HRS, HCC, ACLF, infection, SBP, other liver failure, and laboratory test (Hb, WBC, PLT, TBIL, ALT, AST, ALB, ALP, Scr, PT, INR, Na, K), and treatment data (liver transplantation, hemodialysis, broad-spectrum antibiotic antibiotics, vasoactive drugs, and human albumin). Additionally, for the ICU sub-cohort, matching factors also included vital signs (heart rate, RR, SpO_2_, anion gap, blood pressure, and temperature), ventilation, and use of rifaximin before ICU admission. Crude and adjusted Cox regression analyses were conducted to explore the association between rifaximin and ICU admission and ICU, in-hospital, 90-day, and 180-day death in critically ill patients with liver cirrhosis. Adjusted variables included age and MELD score. Crude and adjusted hazard ratios (HRs) and 95% confidence intervals (CIs) were calculated. Kaplan-Meier curves were constructed to demonstrate the cumulative 90-day and 180-day survival rates and compared by the Log-rank test according to the use of rifaximin. Restricted cubic spline (RCS) regression analysis was performed to explore the association between the total dose of rifaximin and the risk of ICU-admission and ICU, in-hospital, 90-day, and 180-day death. Subgroup analyses were conducted based on gender, hypertension, diabetes, CKD, ascites, HE, GIB, HRS, SBP, HCC, ACLF, and broad-spectrum antibiotic, with adjusted HRs and 95% CIs calculated for each subgroup. A two-tailed *p* < 0.05 was considered statistically significant. All statistical analyses were performed using R (version 4.4.2).

## Figures and Tables

**Figure 1 antibiotics-14-00287-f001:**
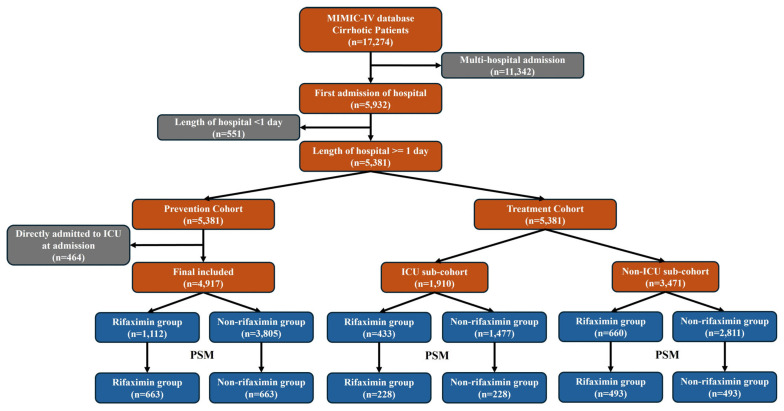
Flowchart of patient selection. ICU, intensive care unit; PSM, propensity score matching.

**Figure 2 antibiotics-14-00287-f002:**
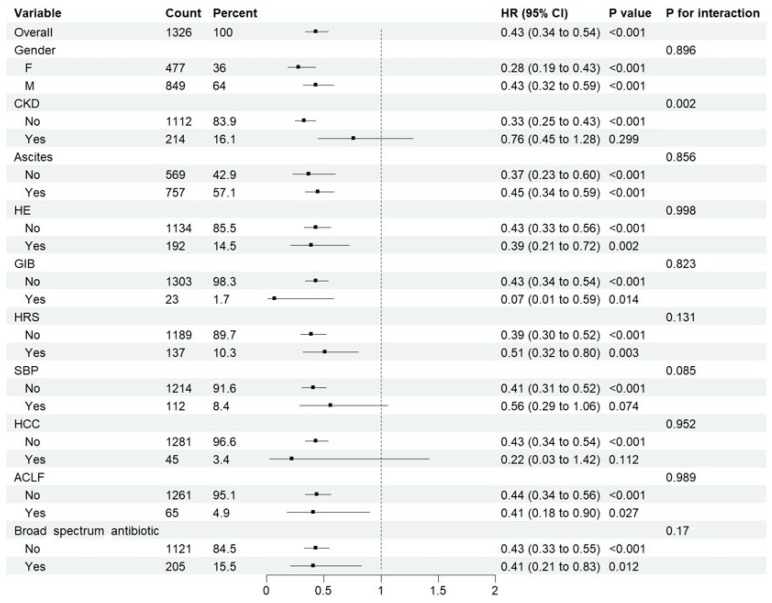
Subgroup analysis of the effect of rifaximin on the risk of ICU admission. F, female; M, male; CKD, chronic kidney disease; HE, hepatic encephalopathy; SBP, spontaneous bacterial peritonitis; HRS, hepatorenal syndrome; ACLF, acute-on-chronic liver failure; GIB, gastrointestinal bleeding; HCC, hepatocellular carcinoma.

**Figure 3 antibiotics-14-00287-f003:**
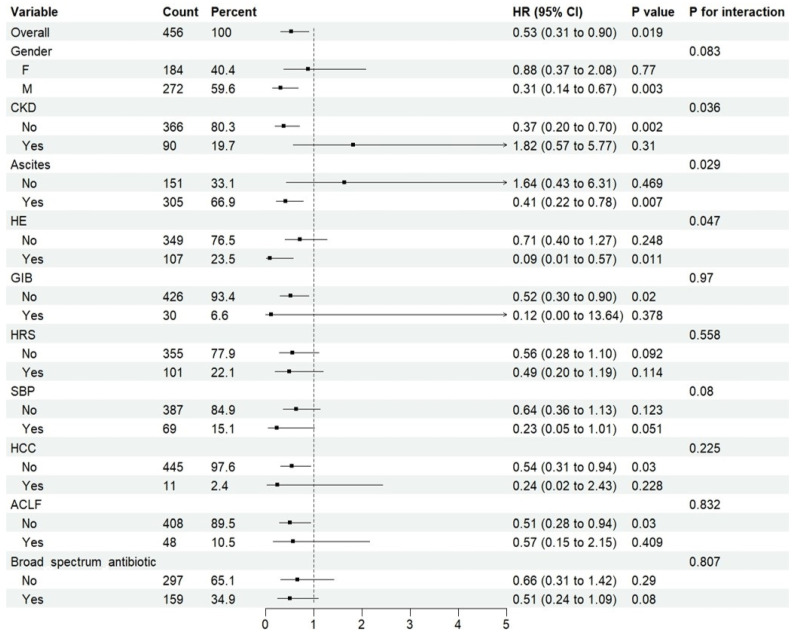
Subgroup analysis of the effect of rifaximin on the risk of ICU death. F, female; M, male; CKD, chronic kidney disease; HE, hepatic encephalopathy; SBP, spontaneous bacterial peritonitis; HRS, hepatorenal syndrome; ACLF, acute-on-chronic liver failure; GIB, gastrointestinal bleeding; HCC, hepatocellular carcinoma.

**Figure 4 antibiotics-14-00287-f004:**
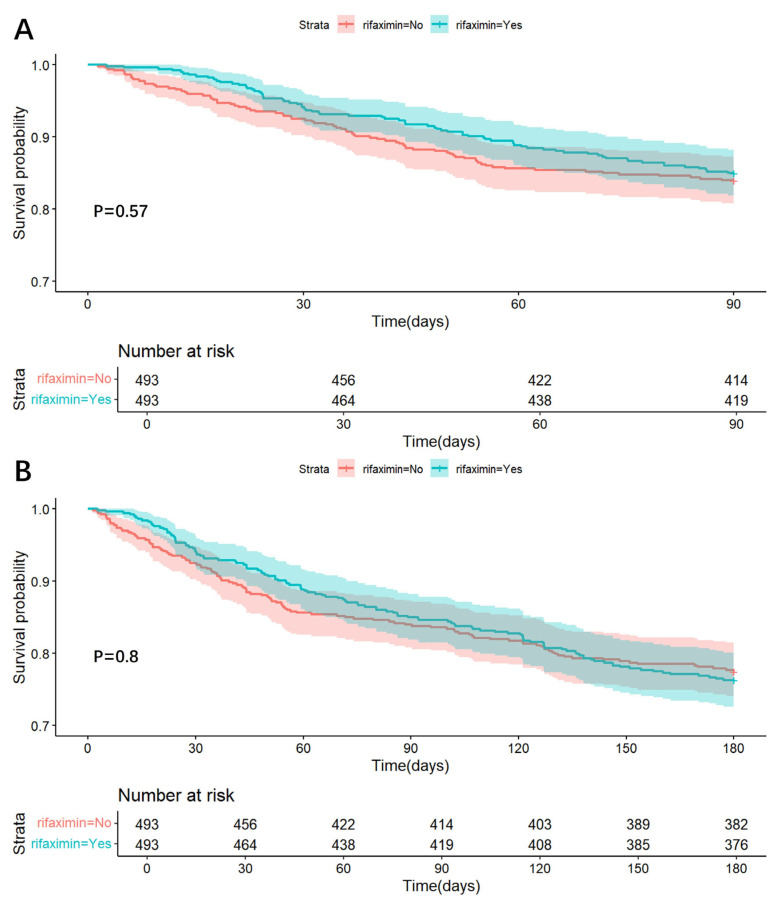
Kaplan-Meier analysis of the effect of rifaximin on 90-day (Panel **A**) and 180-day (Panel **B**) cumulative survival.

**Table 1 antibiotics-14-00287-t001:** Baseline characteristics of included patients.

Variables	Overall (n = 5381)
**Age (years) (median [IQR])**	59.67 [52.28, 68.10]
**Gender (female) (%)**	1894 (35.2)
**Race (%)**	
Asian	173 (3.2)
Black	453 (8.4)
White	3730 (69.3)
Other	1025 (19.0)
**Admission type (%)**	
Elective	78 (1.4)
Observation	916 (17.0)
Surgical	307 (5.7)
Urgent	4080 (75.8)
**MELD score (median [IQR])**	14.74 [10.28, 21.08]
**Hypertension (%)**	1945 (36.1)
**Diabetes (%)**	1665 (30.9)
**CKD (%)**	879 (16.3)
**Infection (%)**	3665 (68.1)
**Etiology of liver cirrhosis**	
Viral hepatitis (%)	1767 (32.8)
Alcohol hepatitis (%)	1336 (24.8)
Autoimmune hepatitis (%)	106 (2.0)
NAFLD (%)	284 (5.3)
**Ascites (%)**	1891 (35.1)
**HE (%)**	378 (7.0)
**GIB (%)**	139 (2.6)
**HRS (%)**	302 (5.6)
**HCC (%)**	130 (2.4)
**ACLF (%)**	146 (2.7)
**SBP (%)**	267 (5.0)
**Other liver failure (%)**	578 (10.7)
**Hb (g/dL) (median [IQR])**	10.70 [9.10, 12.20]
**WBC (10^9^/L) (median [IQR])**	7.10 [4.70, 10.40]
**PLT (10^9^/L) (median [IQR])**	118.00 [75.00, 181.00]
**TBIL (mg/dL) (median [IQR])**	1.60 [0.80, 4.10]
**ALT (U/L) (median [IQR])**	34.00 [21.00, 61.00]
**AST (U/L) (median [IQR])**	61.00 [36.00, 114.00]
**ALB (g/dL) (median [IQR])**	3.10 [2.60, 3.50]
**ALP (U/L) (median [IQR])**	111.00 [78.00, 167.00]
**Scr (mg/dL) (median [IQR])**	0.90 [0.70, 1.40]
**PT (s) (median [IQR])**	15.60 [13.40, 19.30]
**INR (median [IQR])**	1.40 [1.20, 1.80]
**Na (mmol/L) (median [IQR])**	137.00 [134.00, 140.00]
**K (mmol/L) (median [IQR])**	4.00 [3.60, 4.40]
**Ventilation (%)**	470 (8.7)
**Liver transplantation (%)**	26 (0.5)
**Hemodialysis (%)**	194 (3.6)
**Rifaximin (%)**	806 (15.0)
**Monotherapy (%)**	696 (12.9)
**Broad-spectrum antibiotic (%)**	711 (13.2)
**Human albumin (%)**	1114 (20.7)
**Vasoactive agent (%)**	29 (0.5)
**ICU admission (%)**	1910 (35.5)
**In-hospital death (%)**	502 (9.3)

**Abbreviations:** ICU, intensive care unit; IQR, interquartile range; HE, hepatic encephalopathy; SBP, spontaneous bacterial peritonitis; HRS, hepatorenal syndrome; ACLF, acute-on-chronic liver failure; NAFLD, non-alcoholic fatty liver disease; GIB, gastrointestinal bleeding; HCC, hepatocellular carcinoma; Hb, hemoglobin; WBC, white blood cell; PLT, platelet, TBIL, total bilirubin; ALT, alanine aminotransferase; AST, aspartate aminotransferase; ALB, serum albumin; ALP, alkaline phosphatase; Scr, serum creatinine; Na, sodium; K, potassium; PT, prothrombin time; INR, international normalized ratio; MELD, model for end-stage liver disease.

**Table 2 antibiotics-14-00287-t002:** Cox regression analysis between rifaximin and outcomes.

Outcomes	Crude			Adjusted		
	HR	95% CI	*p*	HR	95% CI	*p*
**ICU admission**	0.421	0.334–0.531	<0.001	0.427	0.338–0.539	<0.001
**ICU death**	0.516	0.304–0.876	0.014	0.530	0.311–0.902	0.019
**In-hospital death**	0.141	0.040–0.494	0.002	0.119	0.033–0.429	0.001
**90-day death**	0.912	0.664–1.253	0.571	0.905	0.658–1.245	0.541
**180-day death**	1.035	0.798–1.342	0.796	1.043	0.804–1.353	0.751

**Abbreviations:** ICU, intensive care unit; HR, hazard ratio; CI, confidence interval.

## Data Availability

Data is available on reasonable request.

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
