# Peer review of "Role of Rifaximin in the Prognosis of Critically Ill Patients with Liver Cirrhosis"

_antibiotics, 2025, doi:10.3390/antibiotics14030287_

Round 1
Reviewer 1 Report
Comments and Suggestions for Authors
I have carefully studied the manuscript entitled "Role of rifaximin in the prognosis of critically ill patients with liver cirrhosis" by Bai Z. et al.
The manuscript deals with a very interesting topic, at least to the specialized readership. The authors offer a fresh insight, carrying substantial novelty. The text is well organized, and the language used is devoid of grammatical or syntax errors. However, before considering publication, the authors are kindly invited to discuss / assess the following issues:
Major issues
1) Supplementary Figure 1 (Percent missing values of baseline information at hospital admission): The authors report 93.12% missing data as far as CRP values are concerned; thus, CRP was ommitted from further analysis (as well as γ-GT). Of note, CRP is a crucial parameter for the evaluation of a critically ill patient. Moreover, it is difficult to understand why this detrimental parameter is lacking. The authors are kindly suggested to further discuss / comment on this issue, at least explicitly reporting it as a severe limitation of their study.
2) Figure 4: The Kaplan-Meier analysis of the effect of rifaximin on 90-day (Panel A) and 180-day (Panel B) 256 cumulative survival. Considering that one of the main findings of the study is that rifaximin can significantly improve short-term survival, this should have been reflected in the relevant KM curves. On the contrary, the relevant curves seem to be very close to each other in the first few days (corresponding to "short-term survival"); therefore, a potential difference is debatable and needs fuerthr clarification / explanation. Apart from explicitly elucidating this issue, the authors are kindly requested to rescale the y-axis to ameliorate visulalization.
3) To further consolidate their findings, the authors are kindly suggested to elaborate a Cox regression model with time-dependent covariate to demonstrate that the effect of the predictor (i.e. the use of rifaximin) is not constant over time.
Author Response
Comment 1. The manuscript deals with a very interesting topic, at least to the specialized readership. The authors offer a fresh insight, carrying substantial novelty. The text is well organized, and the language used is devoid of grammatical or syntax errors.
Reply 1. Thank you for your comments.
Comment 2. Supplementary Figure 1 (Percent missing values of baseline information at hospital admission): The authors report 93.12% missing data as far as CRP values are concerned; thus, CRP was ommitted from further analysis (as well as γ-GT). Of note, CRP is a crucial parameter for the evaluation of a critically ill patient. Moreover, it is difficult to understand why this detrimental parameter is lacking. The authors are kindly suggested to further discuss / comment on this issue, at least explicitly reporting it as a severe limitation of their study.
Reply 2. Thank you for your comment. We have checked these data source and confirmed CRP and GGT were missing. According to your comment, we have added this issue to the section of limitation. Please see the words highlighted by yellow in Line 238-240 in the revised manuscript.
Comment 3. Figure 4: The Kaplan-Meier analysis of the effect of rifaximin on 90-day (Panel A) and 180-day (Panel B) 256 cumulative survival. Considering that one of the main findings of the study is that rifaximin can significantly improve short-term survival, this should have been reflected in the relevant KM curves. On the contrary, the relevant curves seem to be very close to each other in the first few days (corresponding to "short-term survival"); therefore, a potential difference is debatable and needs fuerthr clarification / explanation. Apart from explicitly elucidating this issue, the authors are kindly requested to rescale the y-axis to ameliorate visualization.
Reply 3. Thank you for your comment. As well as you pointed out that this potential debatable due to the y-axis range. According to your comment, we rescale the y-axis to ameliorate visualization. Please see the revised Figure 4.
Comment 4. To further consolidate their findings, the authors are kindly suggested to elaborate a Cox regression model with time-dependent covariate to demonstrate that the effect of the predictor (i.e. the use of rifaximin) is not constant over time.
Reply 4. Thank you for your comment. According to your comment, we conducted the time-dependent covariate Cox regression analysis. The time-dependent Cox regression also showed that rifaximin can decrease the risk of ICU admission, ICU death, and in-hospital death, but not in 90-day and 180-day death. The results were summarized as follows.
The Figure and Table please see the attachment.

Reviewer 2 Report
Comments and Suggestions for Authors
The paper entitled:”Role of rifaximin in the prognosis of critically ill patients with liver cirrhosis” by Zhaohui Bai et al. deals with a large (more than 5000 pts) database analysis on critically ill patients with liver cirrhosis from the Medical Information Mart for Intensive Care (MIMIC) IV database including prevention and treatment cohorts exploring 90 day, and 180 day survival rates.
The study presents interesting results: in the prevention cohort, rifaximin could decrease the risk of ICU admission; In the treatment cohort, rifaximin could decrease the risk of ICU and in - hospital death in critically ill patients with liver cirrhosis. Rifaximin can significantly decrease the risk of ICU admission and improve short term survival, it doesn't impact long term survival in critically ill patients with liver cirrhosis.
The underlying hypothesis, that bacterial infections can worsen the prognosis of critically ill cirrotic patients; the study promise to explore whether rifaximin can provide additional therapeutic benefit when combined with broad
spectrum antibiotics, often administered in these patients.
A specific cohort of patients receiving broad spectrum antibiotics has been analysed. Results on this cohort however haven’t apparently been highlighted in results.
The paper therefore seems insubstantial in its ability to examine whether rifaximin can provide additional therapeutic benefit when combined with broad spectrum antibiotics.
We suggest examining the role of antibiotics in all the different cohorts proposed in the study and, more specifically, to separate subgroups characterised by the types of multi-drug therapies used, and maybe the timing of association with rifaximin, in order to extract some possibly useful indication for clinical use.
Tables and Figures are low quality in general, and must all be improved in terms of resolution and readability of the text in them.
At least one explanatory diagram of the underlying biomedical hypothesis could additionally improve the appeal of the paper.
Author Response
Comment 1. A specific cohort of patients receiving broad spectrum antibiotics has been analysed. Results on this cohort however haven’t apparently been highlighted in results. The paper therefore seems insubstantial in its ability to examine whether rifaximin can provide additional therapeutic benefit when combined with broad spectrum antibiotics. We suggest examining the role of antibiotics in all the different cohorts proposed in the study and, more specifically, to separate subgroups characterised by the types of multi-drug therapies used, and maybe the timing of association with rifaximin, in order to extract some possibly useful indication for clinical use.
Reply 1. Thank you for your comment. According to your comment, we analyzed the benefit of rifaximin when combined with different antibiotics (Monotherapy, Gram positive and negative, Gram positive, Gram negative, Anaerobic). Please see as follows.
Additionally, the timing of association with rifaximin cannot be clarified in the current study due to the study design and we have added this issue to the section of limitations. Please see the words highlighted by yellow in Lines 244-245 in the revised manuscript.
Comment 2. Tables and Figures are low quality in general, and must all be improved in terms of resolution and readability of the text in them.
Reply 2. Thank you for your comment. We have improved them and added footnotes. Please see the revised Tables and Figures.
Comment 3. At least one explanatory diagram of the underlying biomedical hypothesis could additionally improve the appeal of the paper.
Reply 3. Thank you for your comment. According to your comment, we have added a graphical abstract in the revised manuscript. Please see as follows.
Figures in the attachment.

Reviewer 3 Report
Comments and Suggestions for Authors
Thank you for inviting me to review this manuscript. It is interesting and informative. I have some comments that could be of use:
- English needs revision
- Figure 2 needs improvement: add a footnote explaining the abbreviations, and remove the ‘_’ symbols. The same stands for Figure 3 and Supplementary Figures 5, 7, and 8
- I wonder if the authors have data regarding the cause of admission to the ICU. If not, I think that it should be added to the limitations subsection of the discussion section
- The research methodology, even though interesting, leaves some issues unexplored. Since this medication does not affect long-term mortality, and we cannot evaluate whether this drug had any adverse effects (especially in doses of >1100mg) due to the nature of the study, it is hard to estimate whether it is a medication worth taking
- Interestingly, the MELD scores of the patients included in the study were <25 (at least the 75% IQR is <25). It is known that there is a paucity of studies regarding rifaximin use in patients with a higher MELD. Are there any data in this subcategory of patients in the present study? If yes, that would be interesting to add in the supplementary section
- Another comment that applies to all tables is that some abbreviations are not explained in the footnote. For example, RR, SOFA, Dbpressure, Sbpressure, SpO2, IQR
English needs revision to increase readability of the manuscript
Author Response
Comment 1. English needs revision
Reply 1. Thank you for your comment. We have improved grammar and syntax. Please see the revised manuscript.
Comment 2. Figure 2 needs improvement: add a footnote explaining the abbreviations, and remove the ‘_’ symbols. The same stands for Figure 3 and Supplementary Figures 5, 7, and 8
Reply 2. Thank you for your comment. We have improved them according to your comments. Please see the revised Figures.
Comment 3. I wonder if the authors have data regarding the cause of admission to the ICU. If not, I think that it should be added to the limitations subsection of the discussion section.
Reply 3. Thank you for your comment. The cause of admission to the ICU cannot be extracted. According to your comment, we have added this issue to the section of limitation. Please see the words highlighted by yellow in Lines 240-241 in the revised manuscript.
Comment 4. The research methodology, even though interesting, leaves some issues unexplored. Since this medication does not affect long-term mortality, and we cannot evaluate whether this drug had any adverse effects (especially in doses of >1100mg) due to the nature of the study, it is hard to estimate whether it is a medication worth taking.
Reply 4. Thank you for your comment. You are right that the efficacy of rifaximin is important for the use of rifaximin and it should be further explored in future research. Therefore, we have added this issue to the section of limitations. Please see the words highlighted by yellow in Lines 241-242 in the revised manuscript.
Comment 5. Interestingly, the MELD scores of the patients included in the study were <25 (at least the 75% IQR is <25). It is known that there is a paucity of studies regarding rifaximin use in patients with a higher MELD. Are there any data in this subcategory of patients in the present study? If yes, that would be interesting to add in the supplementary section
Reply 5. Thank you for your comment. According to your comments, we have explored the role of rifaximin in patients with MELD>25. The results were similar with the whole patients. Please see the following table.
Comment 6. Another comment that applies to all tables is that some abbreviations are not explained in the footnote. For example, RR, SOFA, Dbpressure, Sbpressure, SpO2, IQR
Reply 6. Thank you for your comment. According to your comment, we have re-checked and added the footnotes of all tables and supplementary tables. Please see the words highlighted by yellow in the footnote of each table.
Table in Attachment.

Reviewer 4 Report
Comments and Suggestions for Authors
The article titled: Role of rifaximin in the prognosis of critically ill patients with liver cirrhosis is an extensive retrospective study to estimate the possible role of rifaximine in the outcome of patients with liver cirrhosis. The study design is appropriate, and the detailed description of the different study groups indicated that the patients receiving rifaximin might benefit from avoiding admission to the intensive care unit. They estimated that the subgroup analysis revealed male patients, and those with ascites or HE significantly benefited from rifaximin. Additionally, they found that rifaximin combined with broad-spectrum antibiotics may offer a greater benefit for ICU survival than rifaximin alone, although the difference was not statistically significant.
Some explanation would be necessary to explain their observations on the benefit of dosing of rifaximin, because the usual dosing in the prevention of hepatic encephalopathy is 550 mg BD (https://www.mayoclinic.org/drugs-supplements/rifaximin-oral-route/description/drg-20065817 ). According to the author’s calculation, the benefits of rifaximin began when the total dose exceeded 100 mg, with the optimal dose being 1780 mg. However, when the total dose exceeded 2140 mg, this benefit was lost. The usual daily dose of rifaximin is 1100 mg; their reference is that the optimal dose is estimated as 1780 mg, which indicates only one and half day treatment.
Comments on the Quality of English LanguageI won't comment; it is fine.
Author Response
Comment 1. Some explanation would be necessary to explain their observations on the benefit of dosing of rifaximin, because the usual dosing in the prevention of hepatic encephalopathy is 550 mg BD (https://www.mayoclinic.org/drugs-supplements/rifaximin-oral-route/description/drg-20065817). According to the author’s calculation, the benefits of rifaximin began when the total dose exceeded 100 mg, with the optimal dose being 1780 mg. However, when the total dose exceeded 2140 mg, this benefit was lost. The usual daily dose of rifaximin is 1100 mg; their reference is that the optimal dose is estimated as 1780 mg, which indicates only one and half day treatment.
Reply 1. Thank you for your comments. You are right, the use of rifaximin had been recommended by international guidelines for the secondary prophylaxis in patients with liver cirrhosis and hepatic encephalopathy and the dose was 550 mg BD. Our results showed that when the total dose exceeds 2140mg, the benefit of survival is not significant. The reason might be the related sample is little. During ICU station, only 15 patients received rifaximin exceed 2200mg, Therefore, the role of rifaximin in this dose should be further explored in the future study. We have added this issue to the section of limitations. Please see the words highlighted by yellow in Lines 242-244 in the revised manuscript.
Furthermore, according to your comment, we further explored the association between the dose of rifaximin and its benefit during ICU station in patients with hepatic encephalopathy. Please see the following Figure.
Figure in the attachment.

Round 2
Reviewer 1 Report
Comments and Suggestions for Authors
I have carefully studied the revised version of the manuscript entitled "Role of rifaximin in the prognosis of critically ill patients with liver cirrhosis" by Bai Z. et al.
The authors have adequately responded to all queries raised. The quality of the manuscript has been ameliorated. There are no additional comments / issues raised.
Reviewer 2 Report
Comments and Suggestions for Authors
The paper has been considerably improved in our opinion. It now addresses current study limitations and future perspectives. We think it can be ready for publication.
Reviewer 3 Report
Comments and Suggestions for Authors
The manuscript has been improved